# Light Response and Switching Behavior of Graphene Oxide Membranes Modified with Azobenzene Compounds

**DOI:** 10.3390/membranes12111131

**Published:** 2022-11-11

**Authors:** Ilia Sadilov, Dmitrii Petukhov, Victor Brotsman, Alexandra Chumakova, Artem Eliseev, Andrei Eliseev

**Affiliations:** 1Department of Materials Science, Lomonosov Moscow State University, 1-73 Leninskiye Gory, 119991 Moscow, Russia; 2Department of Chemistry, Lomonosov Moscow State University, 1-3 Leninskiye Gory, 119991 Moscow, Russia; 3European Synchrotron Radiation Facility, 71 Av. des Martyrs, F-38042 Grenoble, France

**Keywords:** graphene oxide, light-responsive membranes, *cis*-*trans* isomerization, water vapor permeance, semi-empirical modelling

## Abstract

Here, we report on the fabrication of light-switchable and light-responsive membranes based on graphene oxide (GO) modified with azobenzene compounds. Azobenzene and *para*-aminoazobenzene were grafted onto graphene oxide layers by covalent attachment/condensation reaction prior to the membranes’ assembly. The modification of GO was proven by the UV-vis, IR, Raman and photoelectron spectroscopy. The membrane’s light-responsive properties were investigated in relation to the permeation of permanent gases and water vapors under UV and IR irradiation. Light irradiation does not influence the permeance of permanent gases, while it strongly affected that of water vapors. Both switching and irradiation-induced water permeance variation is described, and they were attributed to over 20% of the initial permeance. According to in situ diffraction studies, the effect is ascribed to the change to the interlayer distance between the graphene oxide nanoflakes, which increases under UV irradiation to ~1.5 nm while it decreases under IR irradiation to ~0.9 nm at 100% RH. The last part occurs due to the isomerization of grafted azobenzene under UV irradiation, pushing apart the GO layers, as confirmed by semi-empirical modelling.

## 1. Introduction

The variation of the membrane characteristics under external stimuli as attracted progressive attention during the past few years [1,2,3,4]. The interest in it is especially exciting due to the possibility of the precise tuning of the membrane separation performance without process termination occurring or by altering the process conditions. The external stimuli that are used to control the properties of membranes can be classified into three general categories: direct stimuli (pH [5] or a presence specific agents in a feed solution), indirect stimuli (temperature [6] or pressure), and field stimuli (electric [7] or magnetic [8] field, or light irradiation [9,10]). Among others, the last group attracts special attention as it requires neither reagents nor a large amount of energy consumption. Light irradiation can cause a rapid and remote response in the membrane, while it has no effect on the permeating media.

To impart photoresponsive properties to a membrane, photochromic compounds are widely incorporated into the membrane structure. Azobenzene, spiropyran, and their derivatives undergoing reversible *cis*-*trans* isomerization reactions under UV/IR light are commonly utilized in the fabrication of light-responsive membranes [11]. The transition from a straight *trans*-isomer to bended *cis*-form decreases the azobenzene molecule length from ~0.9 to 0.55 nm, thus increasing its dipole moment from 0.5 D to 3.1 D [12]. The approach of the chemical inclusion of azobenzene fragments to the polymer chains or the hybrid frameworks have been successfully utilized for the preparation photoresponsive glassy polymeric membranes and structured MOFs [13,14].

Polymer membranes with bound or physically dissolved azobenzene demonstrate gas permeance variations of up to 29% under irradiation [12,15]. The effect is ascribed to the alterations of the polymer-free volume and matrix polarity during isomerization (due to the difference in the dipole moment of the azobenzene isomers). However, overly flexible polymer chains results in rather limited photoinduced permeation changes. In order to improve the effect, the concept was realized with the inclusion of a photoresponsive group into the rigid structures of the zeolites and MOFs [12,16].

The modification of the zeolites with guest azobenzene compounds enables one to increase the reversible permeability and selectivity variation by over 70 and 50% for the N_2_/CO_2_ and CH_4_/CO_2_ mixtures, respectively [12]. The module structure of the MOFs provides even more opportunities for the fabrication of photoresponsive membranes. Photoresponsive molecules can be incorporated into the structure as a guest compound [17], a linker side chain [16] or a linker backbone [14]. The incorporation of azobenzene into the UiO-67 matrix results in a photoresponsive membrane with an H_2_/CO_2_ selectivity that varies from 10 to 15 [17]. The utilization of photoresponsive side chain linker significantly improves the photoresponsive properties—the UV light irradiation of Cu_2_(AzoBPDC)_2_(AzoBiPyB) MOF provides H_2_/CO_2_ selectivity growth from three to eight with it suppressing the CO_2_ permeance from 10^−7^ to 3 × 10^−8^ mol/(m^2^·Pa·s) [16].

However, the fabrication of switchable defect-free zeolite and MOF membranes still remains a “chemical handicraft” that is mostly associated with the 3-dimentional structure of the MOFs, hampering the preparation of the continuous selective layers, and restricting the shrinkage/expansion without causing selective layer destruction. More stable switchable structures are provided with two-dimensional stacked materials, thus enabling a lattice variation in one dimension. Those are represented by graphene [18], graphene oxides [19], MXenes [20], and the 2D nanoflakes of semiconductors [9], which were successfully employed in hydrogen separation [21,22], air dehumidification [23] and removal of condensable gases [24]. Among the other 2D materials, graphene oxide has gained most of the attention due to its impressive performance in dehumidification processes, its ease of fabrication, and the diverse chemical nature of its functional groups, which are available for chemical grafting [25].

The structure of two-dimensional stacked layers suggests the possibility of there being interlayer gap variation with the absorption of specific guest molecules or permeate ones. In turn, the resulting interlayer distance determines the membrane performance, providing a promising pathway to stimuli-responsive membranes. The external thermal [26], laser [10] and electrical [7] control of the permeation rates was successfully realized in 2D membranes. To strengthen the switching effect, the interlayer space of the two-dimensional membranes was further modified with responsive molecules. Modifications to the graphene oxide interlayer spacing by spiropyran, poly(*N*-isopropylacrylamide) or poly(*N*,*N*-dimethylaminoethyl methacrylate) led to photoresponsive [27], thermoresponsive [6], or CO_2_-responsive membranes. However, photoswitching has not yet being reported in 2D membranes to the best of our knowledge. On the other hand, grafted photosensible azobenzene fragments were already engaged recently for the modification of the graphene oxides to induce thermal energy storage, thus achieving reversible *cis*-*trans* isomerization under illumination, and the relaxation process was slow enough to produce switching behavior [28].

Therefore, in the present study, we propose the use of grafted azobenzene to control the interplane distance in the graphene oxide. We suggest the permeance of the membranes with grafted azobenzene (azo-GO) membranes would be sensitive to UV/IR irradiation due to the photochromic conformational changes of the modifier, thus enforcing an extension of the slits sizes of the GO or by allowing their shrinkage. Moreover, the conformational changes having slow dynamics will enable the switching behavior of the membranes, thus they would not require continuous light exposure. The influence of the *cis*-*trans*-isomerization of the modifiers on the permeance of the azo-GO membranes was revealed with our study, and the variation of the slit size was proven by the direct evaluation of the d-spacing with grazing incidence wide-angle X-ray-scattering.

## 2. Materials and Methods

### 2.1. Preparation and Modification of Graphene Oxide

Graphene oxide was obtained by modified Hummers’ method using the procedure that is described in [23] with a graphite-to-potassium permanganate ratio of 1:20.

The azobenzene grafting onto the medium-flake graphene oxide (MFGO) was carried out by the covalent attachment of diazonium, which was generated in the reaction of aniline hydrochloride (0.65 g, 5 mmol) and sodium nitirite (0.35 g, 5 mmol) in 6 M hydrochloric acid solution. The results of the ^1^H NMR and FT-IR of the extracted benzenediazonium chloride are provided in Appendix A. The mechanism of this reaction was suggested in [28]. The modifier solution was added dropwise to an MFGO suspension in an ammonium buffer (30 mL, 1 g·L^−1^, pH = 9.5) at 0–5 °C. The reaction mixture was stirred at 0–5 °C for 30 min, and then, at an ambient temperature for 3 h.

The grafting of para-aminoazobenzene onto the MFGO nanoflakes was performed using a nucleophilic epoxy ring opening with a formation of 1,2-aninoazobenzene alcohols according to [29]. The competing reaction of amide condensation was limited by there being a very low amount of carboxylic groups in the graphene oxide which was prepared by modified Hummers’ method [23]. The suggested reaction scheme is provided in Appendix A. To induce the reaction, the suspension of MFGO (30 mL, concentration 1 g L^−1^) in a water/ethanol mixture (*v*/*v* = 1/1), and para-aminoazobenzene (100 mg, 0.5 mmol) and 20 mL of 3 M acetic acid were placed in a Teflon autoclave and subjected to a hydrothermal treatment at 70 °C for 24 h.

In both of the cases, the resulting suspensions were separated by filtration using a 0.2-μm membrane filter and successively washed with methanol and deionized water. The filter cakes were dispersed in deionized water and centrifuged at 4400 rpm several times until the pH = 6–7 was reached.

### 2.2. Membranes Preparation

Anodic alumina membranes with average pore diameters of ~40 nm on the top surface and a thickness of ~100 µm were used as supports for the deposition of the selective azo-GO layers. The supports were obtained by the anodic oxidation of aluminum in 0.3 M H_2_C_2_O_4_ at 40 V using a two-stage anodization procedure, which was followed by a residual aluminum etching and pore-opening procedure [30]. The modified graphene oxide suspensions with concentration 1 g·L^−1^ were deposed onto anodic alumina supports by a spin-coating technique under a low vacuum suction (~30 kPa) at 1500 rpm.

### 2.3. Characterization of Composite Membranes

The UV/Vis absorption spectra of the suspensions were recorded using an AvaSpec-2048 fiber optic spectrometer (spectral range 200–1100 nm, optical resolution 2.4 nm, Avantes, Netherlands) equipped with the AvaLight-DH-S-Bal light source (200–2500 nm range, deuterium and halogen lamps, Avantes, Netherlands) at an ambient temperature. An He-Cd laser (325 nm, 20 mW) with a set of neutral density filters was used for the sample exposure procedure.

The qualitative analysis of the functional groups in the GO was performed by FTIR-spectroscopy using IRAffinity-1 spectrometer (Shimadzu, Japan). The spectra were registered in KBr pellets within the wavenumber range of 400–4000 cm^−1^ with a spectral resolution of 2 cm^−1^.

The microstructure of the obtained membranes was characterized by scanning electron microscopy (SEM) using a Nova NanoSEM 230 (FEI Company, Hillsboro, OH, USA) instrument.

The Raman spectra of the azobenzene compounds and azo-GO membranes were acquired using a Renishaw InVia Reflex spectrometer (Renishaw, Gloucestershire, UK) using 325 nm 20 mW He-Cd and 532 50 mW diode lasers excitation. The excitation power and exposure duration was varied in order to reveal the individual *trans*- isomer spectra and minimize the UV exposition if this was necessary.

The permeance of the porous supports, GO, and azo-GO membranes in relation to the permanent gases (H_2_, CH_4_, N_2_, O_2_) were measured using an integral scheme [24]. The feed chamber and calibrated permeate volume were evacuated down to a residual pressure that was lower than 0.1 mbar, the feed side was swept at 1 bar, and the pressure–time dependence in the permeate volume was recorded. The water vapor permeance was measured according to the protocol that was described earlier in [31]. The feed side of the membrane was blown by a gas of controlled humidity, which was obtained by intermixing the dry and humid gas streams. The permeate side of the membrane was swept by Ar. The temperature and humidity of both of the gas fluxes were controlled using HIH-4000 sensors (Honeywell, Guangzhou, China). The influence of the membrane illumination on the membrane permeance and photoswitching behavior was investigated using the custom-designed cell with a quartz window that was located at the feed side. A uniform illumination of the composite membranes was achieved using high-quality focused UV (365 nm, 15 W) and IR (850 nm, 15 W) LEDs. The thermography maps of the membranes under illumination were acquired using a thermal imager, Testo 885-2 (Testo, Titisee-Neustadt, Germany).

The variation of the interlayer distance in the modified GO under UV and IR irradiation was evaluated using the P03 beamline of PETRA III Deutsches Elektronen-Synchrotron (DESY, Hamburg, Germany) in a grazing incidence geometry [19]. An incident photon beam with an energy of 19.8 keV (λ = 0.6262 Å, Δλ/λ = 10^−4^) and a beam size at the sample position of 33 × 27 µm^2^ was used. A grazing incidence angle of 0.4° was selected to maximize the scattering intensity from the samples. The scattered X-rays were captured using Nexus Lambda 9 M detector (pixel size 55 × 55 mm^2^) (DESY, Hamburg, Germany). The sample-to-detector distance was set to SDD1 = 370 mm. A standard calibration of the sample-to-detector distance with silver behenate and lanthanum hexaboride was performed prior to the GIWAXS experiment. High-quality focused UV (365 nm, 15 W) and IR (850 nm, 15 W) LEDs were used for the sample exposure. The diffraction patterns from the modified graphene oxide both in the dark and under UV/IR irradiation were acquired at a controlled humidity. The humidity level was set using flow controllers and monitored using temperature (ds1820, DALLAS-MAXIM, Dallas, TX, USA) and humidity (HIH-4000, Honeywell, China) sensors. The GIWAXS data were analyzed using the DPDAK software package version 1.5.0 (DESY, Hamburg, Germany). The 2D patterns were integrated in an azimuthal angle of 10° around a normal scattering vector (*q_z_*), and the scattering profiles were fitted with Lorenzians (common for X-ray diffraction lineshape).

The conformational changes of the azobenzene fragment in the interlayer space were investigated with semi-empirical modeling. The computations were performed using the unrestricted Hartree–Fock method using PM7 Hamiltonian in the MOPAC2016 software package (Colorado Springs, CO, USA) [23,32]. Graphene oxide with an XPS-derived content of azobenzene, hydroxyl, epoxy and carboxyl groups (0.01 azobenzene, 0.37 C–OH, 0.10 C–O–C, 0.05 C=O molecules per carbon atom) and water content which was derived from the absorption data (0.9 H_2_O molecules per carbon atom [23]) was used in the model. The model was built with a random layout of functional groups in a unit cell containing 100 carbon atoms. To obtain an optimized geometry of the structure, full geometry optimization steps were performed under periodic boundary conditions according to a procedure that is described in [23]. The models, containing various isomers of azobenzene fragments with minimal free energy were used in further evaluations. The isomeric transition was modelled with the forced movement of the last carbon atom in the azobenzene fragment along the z coordinate (0.1 Å Δz step) with a full geometry optimization being performed of the whole structure at each step. Neither the cell parameters nor any over atomic coordinates were fixed during the geometry optimization.

## 3. Results and Discussion

The successful modification of the graphene oxide nanoflakes with azobenzene and para-aminoazobenzene molecules was proved with UV-vis and IR spectroscopy (Figure 1a,b, Appendix A). In addition, X-ray patterns were obtained for pure azobenzene and 4-aminoazobenzene (Appendix A). The absorption spectra of the azobenenzene-modified GO suspension reveals an upshift (~30 nm) of the charge transfer band, which are indicative of the changes in the molecular configuration of azobenzene due to the removal of the second benzene fragment with the condensation of the GO layer (Figure 1a) [28]. On the contrary, only a slight shift of ~3 nm appears in the case of 4-aminoazobenzene-modified GO (Appendix A), which illustrates the preservation of the diazobenzene fragment during the modification procedure. The IR spectroscopy exposed the characteristic N=N vibrations at 1485 cm^−1^ for both of the modifiers (Figure 1b and Appendix A). Similarly, specific C-N (1120–1165 cm^−1^) and N=N (1400–1465 cm^−1^) vibrations appear on the Raman spectra of the azo-GO [28]. A comparison of the IR spectra of the azo-GO membranes and the physical blends of the graphene oxide with the azobenzene compounds (10 wt.%) illustrates both the shift and the broadening of the diazonium peaks, suggesting that there was a chemical interaction between the components (Appendix A). Despite grafting the azobenzene derivatives to the GO layers cannot be unambiguously evidenced with IR, Raman or absorption spectroscopy, and the repeatable extraction of the unbound precursors from the GO colloids does not considerably change the absorption lines intensity after their repeated washing with methanol. The X-ray photoelectron spectroscopy (Phoibos 150 MCD, SPECS, Germany) analysis of the suspensions deposits demonstrate that there is ~2% of nitrogen present in the samples, corresponding to the addition of one azobenzene molecule per ~100 carbon atoms of graphene oxide.

The conformations of azobenzene in the GO structure were studied by Raman scattering and UV-Vis spectroscopy. The UV excitation with the 325 nm He-Cd laser was employed in order to both induce the isomerization and excite the Raman scattering of the samples. According to the Raman studies, the unilluminated azo-GO samples contain at least a portion of trans-isomer in the initial form which can be identified by a characteristic line at 1165 cm^−1^ [33] (Figure 1c). The content of the trans-isomer decreases with the UV irradiation, nevertheless, its marker persists in the spectra of both the individual compound and the azo-GO sample after their prolonged exposure. This suggests that trans-isomers will retain in azo-GO under UV light. It stays in line with that of the common optical absorption spectroscopy, revealing a rather intense trans-isomer absorption band when the sample was under illumination (Figure 1a and Appendix A). A somewhat lower UV impact to the azo-GO reflects, likely, an enhanced trans-isomerization process, which was catalyzed by the graphene oxide [34]. Unfortunately, the cis-isomer marker at 1525 cm^−1^ [33] was not resolved in the Raman spectra even after the prolonged UV exposure due to its possible interference with a wide sp^3^ carbon signal. Thus, according to the spectroscopic studies, a prevailing trans-azobenzene form is anticipated in the equilibrium azo-GO structure, while UV excitation will decrease the trans-form content in the sample. However, these data do not provide any reasonable explanation for the rise of the permeation rate under the UV irradiation condition.

The suspensions were utilized further for the deposition of continuous graphene oxide membranes using anodic alumina substrates with 40 nm pores on the top surface [30]. A spin-coating technique was employed to provide minimum thickness to the gas-tight continuous layers [31]. Thin coatings, which were nearly transparent to 3 keV electrons, formed on the surface during the use of this approach. According to the cross-sectional SEM studies, the thickness of the selective layer of the membrane did not exceeds 170 nm (Figure 1d). According to the gas transport tests, the permeance of the membranes for nitrogen stayed below 1.2 L·m^−2^·bar^−1^·h^−1^, which was 10,000 times lower than the permeance of the AAO support (Appendix A). Nevertheless, the deposited layers exhibits a low-selective Knudsen diffusion trend for the permanent gases, with the permeance being proportional to √M (Figure 1e). On the other hand, the permeance of the water vapors exceeds that of the gases by three orders of magnitude. Such a behavior is characteristic for graphene oxides of various oxidation degrees [23]. At the same time, the fast transport of water molecules depends strongly on the GO slit size and the amount of absorbed water.

The cis-trans isomerization of the grafted molecules was expected to influence the interlayer spacing and the size of the slit permeation channels. Therefore, the permeance of the azo-GO membranes for the water vapors was further investigated under periodic UV/IR light irradiation (Figure 2a). No valuable changes in the permeance of the permanent gases was detected. On the contrary, a well-resolved reversible change of the water transport efficiency was observed with the switching illumination condition (Figure 2a). Pure GO membrane does not respond to UV irradiation, but it exhibits a decrease in the water permeance under IR light possibly due to membrane heating occurring with the corresponding growth of the local water vapors saturation pressure (Appendix A) [9]. The assumption about the local heating of membranes under light irradiation was verified using thermography mapping. UV light heats the membrane to ~0.9 °C, while IR irradiation results in the local heating of it up to 70 °C (Appendix A). Importantly, besides the direct irradiation-induced permeance change, a long-term permeance switching behavior was noticed for the membranes. That manifested in the preservation of water permeance after we switching off the UV/IR light (Figure 2b). When we switched the light off, the permeance slowly returned to its initial values with typical equilibration times being in the order of a few hours. The relaxation periods (fitted with exponential decay function) reveal that there was a half-time of ~2000 s both for azo-GO and aminoazo-GO. The comparable decay constants indicate that there was temperature-induced relaxation in both of the composite membranes. The long-term switching effect reveals the permeance variation that ranges from 9.24 to 10.27 m^3^·m^−2^·bar^−1^·h^−1^ for the azo-GO membrane that was exposed to UV/IR light. A similar behavior was observed for the aminoazobenzene-modified graphene oxide (Appendix A). Despite the permeance changes being too small to be involved directly in practical applications, the principle of the light switching in order to test the membranes’ permeability with the conformational changes of the spacers seems extremely important for further developments of smart membranes.

Importantly, the permeance of the membrane was always increased after the initial UV irradiation, suggesting an expansion of the transport channels. Such an expansion seems to contest the common knowledge of the cis-isomerization of azobenzene under UV-exposure, and therefore, we must reveal both the conformation of the active molecules and the interlayer spacing of GO under illumination.

In order to reveal the nature of the permeance switching effect, we have provided a further investigation of the interlayer spacing in azo-GO that was exposed to UV and IR light during the grazing incidence X-ray scattering. The technique enables is to make a direct evaluation of the interlayer space in situ under the in operando conditions [35]. The initial state of azo-GO significantly differs from the conventional graphene oxides in both dry and 100% relative humidity states, exceeding the typical value of d-spacing, which is over 0.2 nm (Table 1). The expansion of the interlayer spacing corresponds with the intercalation of the azobenzene fragments between the nanoflakes. Indeed, the derived interlayer distance of 1.3 nm corresponds with the sum of the GO layer thickness (~0.6 nm) and the size of trans-azobenzene fragment (0.7 nm). Alongside this, the interlayer distance in the dry state is close to the sum of GO layer thickness and cis-azobenzene transverse projection (~0.35 nm). Importantly, the growth of the interlayer spacing is accompanied by a growing FWHM of the diffraction reflections, revealing that there was a much wider distribution of the interlayer distances. It reflects, obviously, that the nanoflakes corrugations appeared with the grafting of the azobenzene.

The susceptibility of the azo-GO interlayer distance to illumination was found to depend strongly on the ambient conditions. No significant reaction due to the UV/IR irradiation was detected in the dry state. This obviously originates from the insufficient mobility of the azobenzene fragments in a strongly constrained space between the shrunk layers. On the other hand, the membrane becomes light-responsible when it is in humid air. With the UV light illumination, the interlayer spacing of the GO increases to ~1.5 nm, and then, it returns slowly to its initial state when the switching illumination was turned off (Figure 3a,b and Appendix A). An expansion of the interlayer spacing coincides well the membrane permeance rise, but it contradicts the common knowledge of the cis-isomerization of azobenzene under UV illumination as cis-conformation should obviously require much lower interlayer spacing for the grafted molecules (Figure 3c). The expansion of the interlayer space can, however, be explained by fast conformation changes that may have occurred in the grafted azobenzene and the straightening of C–N=N–C fragment in the transitional state [36,37]. We suggest that such conformational changes can impart an internal pressure on the GO layers, thus shifting the equilibrium interlayer distance to the larger values (Figure 3c, state (2)). Contrary to this, IR irradiation does not offer enough energy for the conformational changes while it is heating up the selective layer. It results in the increase in the saturation pressure of the water vapors and a corresponding decrease in the relative partial pressure, P/P_0_. With these conditional changes, the equilibrium absorption capacity of the GO is diminishing, resulting in the shrinkage of the GO structure [23,35]. The decrease in the interlayer distance during the IR exposure evidences that the total energy for that is needed for the conformational changes of the azobenzene fragment is lower when it is compared to the changes of the energy state of graphene oxide.

In order to reveal the cis-trans-isomerization process of azobenzene within the GO lattice, the semi-empirical modelling of the whole structure was performed. The energy profiles, the individual C–N=N, and the dihedral angles were extracted from the models with the forced bending of the azobenzene fragment relative to the GO plane (see experimental section for details). Those profiles are plotted on Figure 3d and Appendix A versus the azobenzene fragment size (the distance of the utmost hydrogen atom from GO plane). Importantly, the most stable configuration reveals the dihedral angle of the C–N=N–C bonds which were ~105°, which can neither be ascribed to pure cis- nor trans-isomers. This angle is characteristic for an excited state of azobenzene [38]. This indicates that the conformation of the bound azobenzene is strongly affected by the interaction with the surrounding media and the graphene oxide layer. Indeed, an analysis of the partial charges of the atoms indicates an excessive charge of ~0.1 $\bar{e}$ on the nitrogen atom that was bound to the benzene ring and ~0.3 $\bar{e}$ on the nitrogen atom that was bound to the graphene oxide layer (compare to ~0.16 $\bar{e}$ on both nitrogen atoms in free azobenzene). Thus, the interaction with the surrounding medium results in the hybridization of the orbitals on the nitrogen atoms, resulting in a strong deviation from the typical geometry of pure sp^2^ cis- and trans-isomer forms of diazobenzene. Numerous modelling efforts with various starting conformations illustrate very closely the resulting minimum free energy of the states that have <5 kcal/mol differences (~0.2 eV or <1 meV/atom). The stabilization of the pure trans-form, with a dihedral angle that is close to 180°, also occurs close to the utmost fragment length, but its free energy remains ~0.6 eV (~3 meV/atom) higher when it is compared to the state of minimal energy. These similar energy values originate likely from an interaction of the azobenzene fragment with the surrounding water molecules and graphene oxide plane.

Importantly, the whole derived energy profile lays within the 90 kcal/mol range, thus corresponding to 3.8 eV (325 nm) of UV exposure. Both switching to the folded conformation (close to *cis-isomer*) and the elongation of the fragment to 0.8 nm can occur within the excitation energy range. The last one corresponds with the straightening of the C-N=N-C fragment under excitation and coincides with the known mechanism for cis-trans isomerization [36,37]. Thus, according to the simulation results, we can consider that an equilibrium length of the azo-GO fragments (~0.65 nm) corresponds closely the trans-isomer, while an expansion of the GO interlayer space under the UV light occurs due to the active transformations of the azobenzene fragments and the appearance of an elongated, excited state (Figure 3b). This state requires the addition of water to be absorbed from the surrounding medium, which is only possible at high humidity levels. Contrary to this, isomerization to a stable *cis*-form does not occurs for grafted azobenzene due to steric hindrances as the excited molecules relax to the minimum energy state that is defined by the surrounding GO and water molecules. According to the model, the folding of azobenzene does not provides strong energy barriers within wide fragment lengths (from ~0.75 nm to ~0.35 nm), thus enabling the soft reduction of the interlayer distance of the GO to ~0.9 nm upon excessive water removal. Therefore, a folded conformation with a ~0.35 nm azobenzene fragment length and with a dihedral angle that is close to 90° is associated with the dry state of azo-GO. These results coincide with the dry state GIWAXS data (Table 1, Figure 3b,c) and suggest that there is a rather a wide range for permeance modulation. Moreover, an extension of the interlayer space in the dry state compared to the pure GO implies the possible modulation of the minimum condensation pressure of the vapors, thus greatly affecting the permeance of the GO membranes [19]. The results are of both experimental and theoretical importance, indicating that there is a strong potential for tuning the permeance and selectivity of graphene oxides membranes by the incorporation of stimuli-responsive modifiers.

## 4. Conclusions

To conclude, the suggested approach of grafting azobenzene compounds onto graphene oxide nanoflakes and azo-GO nanoflakes during their assembly results in the successful fabrication of light-responsive and switchable membranes. The conformational changes of the azobenzene fragments in the interlayer space of the GO under UV light enable the expansion of the slit channels to ~1.5 nm, resulting in the enhancement of their transport through a stacked laminar structure. According to the semi-empirical modeling, the expansion is associated with the excited state of azobenzene. Controversially, the IR light results in the folding of the azobenzene fragments and absorbed water removal upon local heating. The shrinkage of the interlayer distance to ~0.9 nm results in the corresponding permeance diminution. The fast and reproducible variation of the permeance of the water vapors, which were controlled with UV/IR light irradiation which was maintained at 20%, reveals the strong coupling of the stacked nanoflake structure to the conformation of the azobenzene fragments.

Despite the fact that the attained permeance changes were too small for them to be involved directly in practical applications, the principle of the energy-free switching of the permeability of membranes with the conformational changes of spacers seems extremely important for the further development of smart membranes. We believe the efforts on the molecular-scale design of GO membranes with the proper choice of the modifier molecular structure and layers interconnection will enable the widening of the response ranges and achievement of high vapor permeance at low partial pressures. Moreover, the reported approach can easily be extended to various types of staked nanoflake membranes, including graphenes, MXenes, and other 2D compounds, while rich 2D-flake chemistry will provide wide opportunities for the further invention of stimuli-responsive smart membranes for the smart control of dehumidification and vapor separation processes.

## Figures and Tables

**Figure 1 membranes-12-01131-f001:**
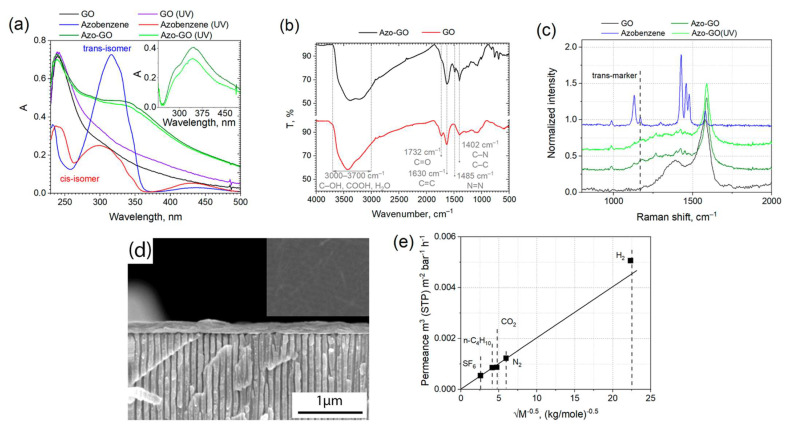
UV-Vis optical absorption spectra of azobenzene, GO, and azo-GO prior and after UV exposure (325 nm, 10 mW) for 20 min (**a**). The difference spectra for azo-GO are provided on the inset. IR spectra of GO and azo-GO samples (**b**). The spectra of pure azobenzene compounds and their physical mixtures with GO are provided in Appendix A. Raman spectra for azo-GO membrane collected prior and after UV exposure (325 nm, 10 mW) for 20 min (**c**). The spectra of azobenzene and pure graphene oxide obtained in the same conditions are provided for reference. The position of trans-azobenzene marker is given in [33] (c). Cross-sectional and top-view (inset) scanning electron microscopy images of Azo-GO composite membrane (**d**). The inset has the same magnification. Gas permeance of Azo-GO composite membrane vs. reverse square root of molecular weight, indicating Knudsen diffusion mechanism of the permeance of the gases (**e**).

**Figure 2 membranes-12-01131-f002:**
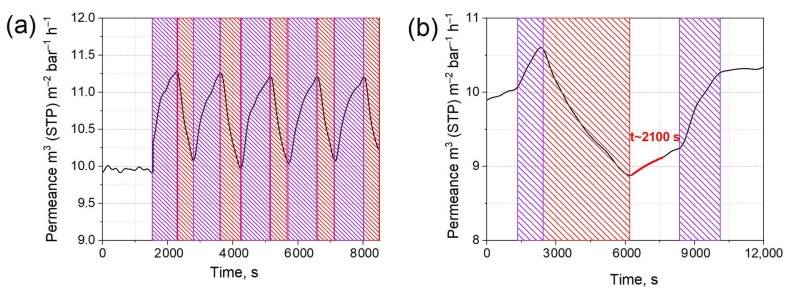
Water vapor permeance of azo-GO membrane subjected to UV (365 nm)/IR (850 nm) irradiation (**a**,**b**). The measurements were performed at feed stream relative humidity of 75%, 1 bar and 25 °C. Violet shaded areas indicate UV light irradiation periods; red shaded areas correspond illumination of membrane with IR light.

**Figure 3 membranes-12-01131-f003:**
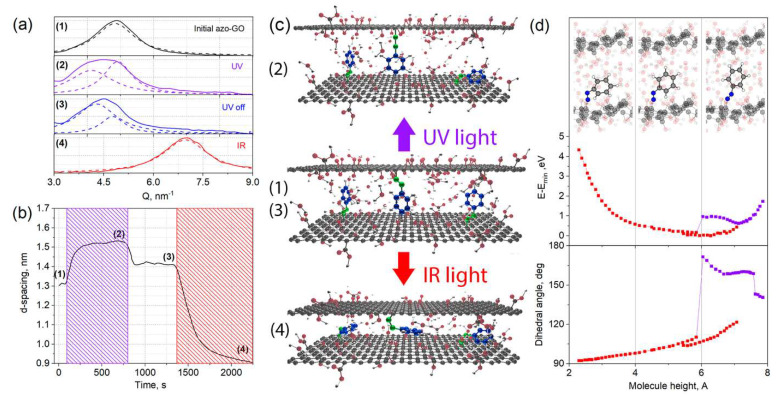
Experimental grazing incidence X-ray scattering data (**a**) for initial azo-GO membrane (1), the membrane exposed to UV (365 nm) illumination (2), after relaxation in the dark (3) and the membrane exposed to IR (850 nm) illumination (4). In situ variation of interlayer distance for azo-GO membrane under UV/IR exposure (**b**). Violet shaded areas indicate UV light exposure; red shaded areas correspond illumination of membrane with IR light. The measurements were performed at relative humidity of ~100%, 1 bar and 25 °C. Schematic representation of switching behavior of azo-GO membrane structure revealing the changes of interlayer spacing with cis-trans-isomerization of azobenzene fragments between graphene oxide nanoflakes (**c**). The attribution of the structures is given on (**a**–**c**) with numbers 1–4. The energy profiles and dihedral C–N=N–C angles for azobenzene fragment according to semi-empirical modelling of azo-GO structure (**d**). The corresponding conformations of the fragment are depicted on the top of (**d**). Excessive water-filled space coincides an ability of GO layers shrinkage with decrease of water chemical potential (humidity).

**Table 1 membranes-12-01131-t001:** GIWAXS data for interlayer spacing and interlayer spacing dispersion in radial and azimuthal direction for pure graphene oxide and azo-GO membranes.

Sample	Humidity, %	D-Spacing, nm	D-Spacing RSD (Radial FWHM of Reflection), nm^−1^	Nanoflakes FWHM. χ
GO	0	0.722	0.909	18.35
GO-azo	0	0.866	1.030	11.37
GO	100	1.153	0.659	11.73
GO-azo	100	1.300	1.433	7.61

## Data Availability

The data supporting the findings of this study are presented in the Supporting information file and also are available upon reasonable request.

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
