# Peer review of "Light Response and Switching Behavior of Graphene Oxide Membranes Modified with Azobenzene Compounds"

_membranes, 2022, doi:10.3390/membranes12111131_

Round 1
Reviewer 1 Report
This paper reports on the permeance behavior of azobenzene-grafted graphene oxide (azo-GO) for water vapor. The vapor permeance increased under UV illumination, while decreased under IR irradiation, only when the humidity is enough high. The mechanism was investigated by X-ray diffraction and semi-empirical modeling, revealing that the increased interlayer distance under UV originates from the strengthening of C-N=N-C, while the decreased interlayer distance under IR originates from the local heating. The authors' explanation seems almost consistent with the results, but some concerns should be resolved.
The major concern is about heating effect versus isomerization effect. Not only IR irradiation, UV irradiation also increases temperature due to the photothermal effect. The authors should estimate the extent of the effect or measure the temperature change by thermography.
The returning behavior after light off seems related directly to the thermal relaxation property of azobenzene. Is there any difference between pure azobenzene-grafted GO and 4-aminoazobenzene-grafted GO?
What is the state of azobenzene in azo-GO in the dry state? The size (0.866 nm) in the dry condition is much smaller than the size (1.300 nm) in the humid condition.
The following are minor points I noticed:
The expression "energy-free switching" seems a bit confusing in the context of this paper.
Does dotted lines in Figure 3a represent gaussian fitting?
At line 202, Figure1c should be corrected to Figure 1d.
At line 206, Figure1d should be corrected to Figure 1e.
At line 256, Figure 2c should be corrected to Figure 1c.
Author Response
Reviewer #1:
Dear Reviewer! We are glad to hear your opinion about the creativity of our manuscript. According to your suggestions, we have introduced a number of corrections to the manuscript. Our reply and the list of changes is given below.
- The major concern is about heating effect versus isomerization effect. Not only IR irradiation, UV irradiation also increases temperature due to the photothermal effect. The authors should estimate the extent of the effect or measure the temperature change by thermography.
AUTHORS
Temperature of the membrane was initially monitored in the in-situ diffraction experiments with IR-sensor. Those measurements were taken into account when describing the photoswitching and temperature-induced changes in the interlayer spacing of GO membranes. However, due to strong uncertainty in the temperature measurements by this method (for small-sized membranes), we avoided releasing the data in the original manuscript. Now we have complemented the data with IR thermography maps of the membranes placed in the measurement cell using Testo 885 thermal imaging camera. The membrane heat up of ~0.9 °C was detected under UV light with the method, while in case of IR light, the maximum membrane temperature increases up to 70 °C. Those results agree well with the permeance of pure GO, being closely irresponsible to UV-light, but diminishing strongly under IR-light due to an increase in the equilibrium pressure of saturated water vapors upon heating. The thermal images are given below.
ACTION TAKEN
IR thermography maps under UV and IR light were added to the supplementary materials. A comment on the temperature measurements was added to the main text of the manuscript.
- The returning behavior after light off seems related directly to the thermal relaxation property of azobenzene. Is there any difference between pure azobenzene-grafted GO and 4-aminoazobenzene-grafted GO?
AUTHORS
No significant quantitative difference was detected in relaxation behavior of azobenzene-grafted GO and 4-aminoazobenzene-grafted GO (compare Figure 2b and Figure S6b). The relaxation rates of the permeance after IR-off, fitted with indicate rather close preexponential and exponential factors, revealing possible temperature-induced relaxation in both the composite membranes. With a limited kinetic data available, we prefer to avoid speculating on the topic.
ACTION TAKEN
A short comment on close relaxation rates has been added to the supplementary information and the manuscript text.
- What is the state of azobenzene in azo-GO in the dry state? The size (0.866 nm) in the dry condition is much smaller than the size (1.300 nm) in the humid condition.
AUTHORS
According to the results of geometry optimization, “dry state” interlayer distance can only correspond to a bended structure of azobenzene with dihedral C-N=N-C angle slightly above 90° (see insets on fig.3d). It does not correspond neither cis- nor trans-configuration, having an excessive energy of ~0,5 eV per molecule. As soon as pure cis-isomer cannot appear in the grafted form due to steric hindrances, this bended configuration is considered as a minimum energy state for dry GO (as soon as increasing d-spacing of GO in dry conditions also carries an excessive energy as it was revealed in Ref. [23] given in manuscript).
ACTION TAKEN
Clarifying remarks has been added to Fig 3d caption and manuscript text to underline possible azobenzene configurations in dry and wet states.
- The expression "energy-free switching" seems a bit confusing in the context of this paper.
ACTION TAKEN
Corrected
- Does dotted lines in Figure 3a represent gaussian fitting?
AUTHORS
The dotted lines represent Lorenz fits (common for X-ray diffraction lineshape).
ACTION TAKEN
A comment on the lineshape used for fitting of X-ray scattering data have been added to the experimental section.
6.At line 202, Figure1c should be corrected to Figure 1d. At line 206, Figure1d should be corrected to Figure 1e. At line 256, Figure 2c should be corrected to Figure 1c.
ACTION TAKEN
Thanks. Corrected.
Reference
- Chernova, E.A.; Petukhov, D.I.; Chumakov, A.P.; Kirianova, A. v; Sadilov, I.S.; Kapitanova, O.O.; Boytsova, O. V.; Valeev, R.G.; Roth, S. V.; Eliseev, A.A.; et al. The Role of Oxidation Level in Mass-Transport Properties and Dehumidification Performance of Graphene Oxide Membranes. Carbon N Y 2021, 183, 404–414, doi: 10.1016/j.carbon.2021.07.011.

Reviewer 2 Report
The authors made membranes of graphene oxide (GO) modified with azobenzene and systematically clarified the light response and switching of the water vapor permeance. While the light-induced reversible cis-trans transition in the azobenzene/GO system was already demonstrated in ref [27], as the energy storage, the applications of this mechanism for the water vapor controllable membranes provide new insight for applications, such as building materials responsible for sunlight. Thus it can evoke further applied research. Therefore, I think this manuscript is good to publish in 'membranes'. However, before publishing the manuscript, the author should recheck minor errors. For example, in line 206, Fig. 1d is Fig. 1e, and in line 255, Raman spectra are not included in Fig. 2.
Author Response
Reviewer #2:
Dear Reviewer! Thank you for attention to our manuscript. We appreciate your support on publication of our research in “Membranes”. According to your remarks, we have corrected the noted errors and spell-checked our manuscript once again.
- For example, in line 206, Fig. 1d is Fig. 1e, and in line 255, Raman spectra are not included in Fig. 2.
ACTION TAKEN
Corrected. The text of the manuscript was checked for spelling and grammar and a number of errors was corrected.

Reviewer 3 Report
The authors report an azobenzene grafted graphene oxide membrane with switchable permeance of water vapors via light stimulation. There is no doubt that this paper has many new discoveries, but some problems need to be revised. The author should make further revision to the article.
General comments:
1. The reason choosing graphene oxide as the modified material should be introduced in Introduction. Does graphene oxide contribute to the light-response and switching behavior?
2. The FTIR and 1H NMR of the diazonium compound should be provided to analyse the preparation and modification of graphene oxide.
3. The mechanism diagram of the reaction between diazonium and MFGO should be provided.
4. Line 107-108, “Grafting of para-aminoazobenzene onto MFGO nanoflakes was realized through imine and amide condensation”, this description is not accurate, so the condensation should be between amino group form aminoazobenzene and carboxylic acid from GO.
5. In the Figure 1a, the UV-vis spectra of para-aminoazobenzene and the corresponding modified GO are not presented, so it is unable to get the conclusion that "Both the molecules reveal the upshift (~30 nm) of the charge transfer peak in the 185 modified GO suspensions". The UV-vis spectrum of para-aminoazobenzene and the corresponding modified GO need to be supplied. Besides the wavelength rang between 280 nm and 350 nm need to be magnified in order to see the absorption change clearly.
6. For the IR, the corresponding molecules of azo and aminoazo should be presented for comparison. If the condensation reaction happened, the absorption of amide group should be able to see. But according to the IR, no obvious change of the absorption can be seen. The authors need to give a deep discussion about it or give more strong evidence to prove that the GO has been modified with the azo or aminoazo. Besides, the physical blended azo molecules and GO mixture as the control sample should be provided.
7. According to the Figure 1, the SEM is Figure 1d instead of Figure 1c.
8. The description of line 202-204 "According to gas transport tests, the permeance of the membranes for nitrogen lays below 1.2 m-2·bar-1·h-1, exceeding flow resistance of AAO supports over 10000 times" is not correct. According to Figure 1e, it shows that the permeance of the membranes for nitrogen lays is around 0.001 m3·m-2·bar-1·h-1 instead of 1.2 m3·m-2·bar-1·h-1.
9. There is no Figure 2c in the Figure 2. So the corresponding description is unable to evaluate.

Author Response
Reviewer #3:
Dear Reviewer! Thank you for close attention to our manuscript. According to your comments, we performed additional experiments to increase the investigative depth of our work. Furthermore, with the new data we introduced a number of corrections to the manuscript, those we believe improved its quality. Point-by-point answers to your questions are given below.
- The reason choosing graphene oxide as the modified material should be introduced in Introduction. Does graphene oxide contribute to the light-response and switching behavior?
AUTHORS
The choice of graphene oxide was dictated by the ease of fabrication of large single-layer graphene oxide nanoflakes and ease of deposition of ultrathin defect-free GO membranes. Due to these factors GO is considered as one of most promising materials in modern membrane technology.
Graphene oxide itself does not contributes switching behavior under UV, as revealing no conformational changes. Nevertheless, the interlayer spacing in GO is highly sensible to water chemical potential, which is in turn governed by both partial water vapor pressure and temperature. Therefore GO becomes responsible to heating (including IR), due to decreasing P/P0 (the ratio of partial water vapors pressure to saturated vapors pressure). Shrinking the interlayer gap leads to diminution of the permeance through the membrane. The effect for pure GO is well illustrated in supplementary information S6.
ACTION TAKEN
The reason for choosing graphene oxide as the modified material has been added to the introduction section. A clarifying remark on the IR and heat responsive behavior of GO has been added to Results and Discussion.
- The FTIR and 1H NMR of the diazonium compound should be provided to analyse the preparation and modification of graphene oxide.
AUTHORS
According to your suggestion, we have accomplished 1H NMR and FTIR studies of the diazonium compound. The results are provided below.
Benzenediazonium chloride. 1H NMR (600 MHz, D2O, ppm): δH 8.61 (d, 3JHH 8.1 Hz, 2H), 8.31 (t, 3JHH 7.3 Hz, 1H), 8.09 (t, 3JHH 8.2 Hz, 2H).
1H NMR spectrum of benzenediazonium chloride in D2O (Figure S1a) reveals one doublet at 8.61 ppm with spin-spin coupling constant 3JFF = 8.1 Hz and two triplets at 8.31 (3JHH 7.3 Hz) and 8.09 (3JHH 8.2 Hz) ppm attributable to phenyl protons. FT-IR spectrum of benzenediazonium chloride (Figure S1b) in the solid state (KBr pellet) containing the νN≡Nstretching frequency at 2295 cm−1 is in good agreement with the literature data [1].
ACTION TAKEN
1H NMR and FTIR data of the diazonium compound have been added to supplementary information.
- The mechanism diagram of the reaction between diazonium and MFGO should be provided.
AUTHORS
In our study we rely upon the grafting protocol reported in Ref [2]. According to the initial mechanism, reported by the authors, grafting of azobenzene groups onto GO nanoflakes is realized by coupling reaction between benzenediazonium chloride, which behaves as a weak electrophile and attacks carbon atoms with high electron density in phenol ring of graphene oxide. With rather a scarce data on the structure of the product, we avoid speculating on the mechanism of reaction, restricting ourselves with referencing to the published data.
On the other hand, grafting of azobenzene to GO flakes is indirectly supported with our diffraction data, revealing extension of GO nanoflakes by over 0.2 nm under UV-light, as unbound azobenzene compounds can unlikely wedge nanoflakes away with cis-trans-isomerization.
ACTION TAKEN
A remark on published mechanism of the reaction has been added.
- Line 107-108, “Grafting of para-aminoazobenzene onto MFGO nanoflakes was realized through imine and amide condensation”, this description is not accurate, so the condensation should be between amino group form aminoazobenzene and carboxylic acid from GO.
AUTHORS
We consider both imine (with carbonyl or epoxy-group) and amide (with carboxylic acid) condensation reactions are possible between aminoazobenzene and GO. On the other hand, according to XPS data [3], a minimum amount of carboxylic-groups is available in GO samples obtained by modified Hummer’s method. Therefore, amide condensation with para-aninoazobenzene unlikely occurs. We believe that due to high reactivity of GO epoxide groups the reaction between GO and para-aninoazobenzene mostly proceeds via nucleophilic epoxy ring opening with formation of 1,2-aninoazobenzene alcohols on the GO nanoflakes (Figure S2). Similar reactions were reported in ref. [3].
|
|
ACTION TAKEN
The corresponding discussion on condensation pathways is added to the manuscript text. The possible grafting mechanism is added to supplementary information.
- In the Figure 1a, the UV-vis spectra of para-aminoazobenzene and the corresponding modified GO are not presented, so it is unable to get the conclusion that "Both the molecules reveal the upshift (~30 nm) of the charge transfer peak in the 185 modified GO suspensions". The UV-vis spectrum of para-aminoazobenzene and the corresponding modified GO need to be supplied. Besides the wavelength rang between 280 nm and 350 nm need to be magnified in order to see the absorption change clearly.
AUTHORS
We gratefully acknowledge Reviewer for this comment, allowing to correct an erroneous statement in our manuscript. While an absorption spectrum of azobenzene modified GO indeed indicates ~30 nm shift compared to diazobenzene compound due to removal of the second benzene fragment, the same is not true in the case of 4-aminoazobenzene grafting. Only a slight shift is observed in the second case (see below). We introduced the corresponding corrections to the manuscript providing the required data in the supplementary information.
ACTION TAKEN
The corresponding discussion has been corrected and the spectra of 4-aminoazobenzene modified GO were added to supplementary information. The wavelength range on the Figure 1a was updated.
- For the IR, the corresponding molecules of azo and aminoazo should be presented for comparison. If the condensation reaction happened, the absorption of amide group should be able to see. But according to the IR, no obvious change of the absorption can be seen. The authors need to give a deep discussion about it or give more strong evidence to prove that the GO has been modified with the azo or aminoazo. Besides, the physical blended azo molecules and GO mixture as the control sample should be provided.
AUTHORS
We provide all the IR spectral data for initial components, our composite membranes and the control physical mixtures (given below). It illustrates both the shift and broadening of peaks, corresponding to azobenzene compounds in the composite membranes. However, we can hardly unambiguously ascribe one of appearing signals to amide group without deep studies/modeling. Despite resolving significant spectral changes, with the data, we still cannot unambiguously state grafting of the modifiers to GO nanoflakes. With all the data on UV-Vis, IR and permeation we can only suggest chemical binding to nanoflakes. Therefore, while adding a discussion on the new IR data, we left the original formulations in the discussion on possible grafting.
|
|
|
Figure S4. FT-IR (KBr) spectra of azo-GO and aminoazo-GO membranes in comparison with initial MFGO, initial azobenzene compounds and their control physical mixtures. The spectra reveal both shifting and broadening of azobenzene components signals in the membranes.
ACTION TAKEN
The required spectra have been added to supplementary information. A short discussion on the spectral changes upon GO modification has been included to discussion.
- According to the Figure 1, the SEM is Figure 1d instead of Figure 1c.
ACTION TAKEN
Corrected
- The description of line 202-204 "According to gas transport tests, the permeance of the membranes for nitrogen lays below 1.2 m-2·bar-1·h-1, exceeding flow resistance of AAO supports over 10000 times" is not correct. According to Figure 1e, it shows that the permeance of the membranes for nitrogen lays is around 0.001 m3·m-2·bar-1·h-1 instead of 1.2 m3·m-2·bar-1·h-1.
ACTION TAKEN
Should be 1.2 l·m-2·bar-1·h-1 (liters), which is 1000 less that m3·m-2·bar-1·h-1. Corrected.
- There is no Figure 2c in the Figure 2. So the corresponding description is unable to evaluate.
ACTION TAKEN
Corrected.
References
- Nuttall, R.H.; Roberts, E.R.; Sharp, D.W.A. The Infra-Red Spectra of Some Stable Diazonium Salts. Spectrochimica Acta1961, 17, 947–952, doi:10.1016/0371-1951(61)80030-1.
- Pang, W.; Xue, J.; Pang, H. A High Energy Density Azobenzene/Graphene Oxide Hybrid with Weak Nonbonding Interactions for Solar Thermal Storage. Sci Rep 2019, 9, 5224, doi:10.1038/s41598-019-41563-w.
- Vacchi, I.A.; Spinato, C.; Raya, J.; Bianco, A.; Ménard-Moyon, C. Chemical Reactivity of Graphene Oxide towards Amines Elucidated by Solid-State NMR. Nanoscale 2016, 8, 13714–13721, doi:10.1039/C6NR03846H.

Round 2
Reviewer 1 Report
The revised manuscript seems to have resolved all my concerns.
Author Response
Reviewer #1:
Dear Reviewer! Thank you for your work! We are glad to hear that the revised manuscript seems to have resolved all your concerns.

Reviewer 3 Report
This manuscript was revised according to the comments, and the quality is much improved. I recommend its publication after minor revision.
1.Please show the diffraction data of the control sample (pure azobenzen compound) to compare with that of azo-GO.
Author Response
Reviewer #3:
Dear Reviewer! We are glad to hear that the quality of the manuscript is much improved. We perform XRD analysis of control samples azobenzene and p-aminoazobenzene and add the information into the supplementary materials. We believe that this allows us to improve the manuscript quality
- Please show the diffraction data of the control sample (pure azobenzene compound) to compare with that of azo-GO
AUTHORS
We perform XRD analysis of control samples pure azobenzene and p-aminoazobenzene. The obtained patterns were compared with the diffraction pattern in The Cambridge Crystallographic Data Centre database. The diffraction pattern of p-aminoazobenzene corresponds well to the pattern given in the CDCC database, while the pattern for azobenzene correspond to a mixture of cis- and trans isomers of azobenzene. Also, for comparison, we added the diffraction pattern of the dry azo-GO membrane. On the diffraction pattern of the azo-GO membrane, one can observe the absence of peaks corresponding to azobenzene. This can be explained by the low quantity of immobilized molecules and another ordering of these molecules on graphene oxide nanosheets compared with a bulk phase of azobenzene.
ACTION TAKEN
The diffraction data were added to the supporting information as Figure S5.
